# Adapter Chimeric Antigen Receptor (AdCAR)-Engineered NK-92 Cells for the Multiplex Targeting of Bone Metastases

**DOI:** 10.3390/cancers13051124

**Published:** 2021-03-05

**Authors:** Stefan Grote, Frank Traub, Joerg Mittelstaet, Christian Seitz, Andrew Kaiser, Rupert Handgretinger, Sabine Schleicher

**Affiliations:** 1Department of Hematology and Oncology, University Hospital Tuebingen, Children’s Hospital, 72076 Tuebingen, Germany; christian.seitz@med.uni-tuebingen.de (C.S.); rupert.handgretinger@med.uni-tuebingen.de (R.H.); sabine.schleicher@med.uni-tuebingen.de (S.S.); 2Department of Orthopedics and Traumatology, University Medical Center Mainz, Johannes Gutenberg University Mainz, 55131 Mainz, Germany; frank.traub@unimedizin-mainz.de; 3Miltenyi Biotec GmbH, 51429 Bergisch Gladbach, Germany; joergmi@miltenyi.com (J.M.); andrewk@miltenyi.com (A.K.)

**Keywords:** NK-92, adapter, chimeric antigen receptor, bone metastasis, solid tumors

## Abstract

**Simple Summary:**

Metastatic disease remains one of the biggest challenges for tumor therapy. The aim of our study was the preclinical evaluation of adapter chimeric antigen receptor (AdCAR)-engineered NK-92 cell efficacy as a possible treatment strategy for various types of bone metastatic cancers. We confirmed that AdCAR NK-92 cells successfully induces tumor cell lysis in bone metastasis cell lines derived from mammary, renal cell and colorectal carcinoma as well as melanoma in a specific and controllable manner, thus, establishing a potent cellular product with universal applicability and quick clinical translation potential for the treatment of solid tumors, including metastases.

**Abstract:**

Background: Since metastatic spreading of solid tumor cells often leads to a fatal outcome for most cancer patients, new approaches for patient-individualized, targeted immunotherapy are urgently needed. Methods: Here, we established cell lines from four bone metastases of different tumor entities. We assessed AdCAR NK-92-mediated cytotoxicity in vitro in standard cytotoxicity assays as well as 3D spheroid models Results: AdCAR-engineered NK-92 cells successfully demonstrated distinct and specific cytotoxic potential targeting different tumor antigens expressed on cell lines established from bone metastases of mammary, renal cell and colorectal carcinoma as well as melanomas. In that process AdCAR NK-92 cells produced a multitude of NK effector molecules as well as pro inflammatory cytokines. Furthermore, AdCAR NK-92 showed increased cytotoxicity in 3D spheroid models which can recapitulate in vivo architecture, thereby bridging the gap between in vitro and in vivo models. Conclusions: AdCAR NK-92 cells may provide an interesting and promising “off-the-shelf” cellular product for the targeted therapy of cancers metastasizing to the bone, while utilization of clinically approved, therapeutic antibodies, as exchangeable adapter molecules can facilitate quick clinical translation.

## 1. Introduction

Cancer is one of the leading causes of death worldwide [1]. Despite the development of new therapeutic approaches and significant improvement of survival rates in the last 20 years, metastatic disease, primarily to the bone, lungs and brain, remains incurable and is the main cause of cancer-associated mortality [2]. Bone is the third most common site for tumor metastasis after the lungs and the liver [3,4]. Depending on the origin of the primary tumor, bone metastases are diagnosed in approximately 75% of patients with breast or prostate cancer and to a lesser extent in other cancers, including lung, kidney, liver and melanoma [5].

The most frequent site affected by bone metastasis is spine, including thoracic spine (63.6%) and lumbar spine (53.8%), followed by ribs (57.5%), pelvis (54.1%), sternum (44.3%), scapula (25.1%), and femur (24.8%) [6]. Bone metastases are often associated with hypercalcemia, severe bone pain, pathological fractures and spinal cord compression, leading to increased morbidity in cancer patients [7].

The treatment of bone metastases in patients with solid tumors is generally palliative, with very limited opportunities for complete eradication. Given the limited success of standard therapies at preventing or treating bone metastatic cancer novel therapeutic strategies designed to destroy dormant disseminated tumor cells and existing cancer metastases is an objective of paramount importance. Adoptive immunotherapy may represent such an innovative treatment option for bone metastases but has not been assessed in detail yet.

Chimeric antigen receptor (CAR)-modified lymphocytes represent a promising immunotherapeutic approach that involves the genetic modification of immune cells to express synthetic recombinant receptors on the cell surface, leading to predefined target specificity [8]. The CAR fusion protein typically comprises an extracellular single-chain variable fragment (scFv) of an antibody for target recognition, a hinge region to provide flexibility, a transmembrane region, and an intracellular activation domain for signal transduction. The core component of the CAR endodomain contains either the CD3ζ portion of the TCR complex or the γ-chain of the high-affinity IgE Fc receptor (first generation CAR), whereas the addition of one or two costimulatory domains derived from CD28, 4-1BB, OX40, ICOS or CD27 for example resulted in second and third generation CAR T cells with sustained activation, persistence and improved functions. Upon expression in lymphocytes, the CAR can engage its target antigen and thereby activating a variety of effector responses resulting in targeted cell killing [9].

Immunotherapy using autologous CD19 CAR T cells has resulted in impressive clinical response rates in patients with relapsed or refractory B cell malignancies [10,11,12]. Recently, the US Food and Drug Administration (FDA) and the European Medicines Agency (EMA) approved two CD19 CAR T cell therapeutics, Kymriah (Tisagenlecleucel) and Yescarta (Axicabtagene ciloleucel), for patients with acute lymphoblastic leukemia and certain types of relapsed or refractory large B cell lymphoma. However, the patient-specific nature of this cell therapy, complex manufacturing workflows and the substantial risk of severe side effects, including cytokine release syndrome (CRS) and immune effector cell–associated neurotoxicity syndrome (ICANS) have led to concerns over costs and safety [13]. In addition, difficulties in obtaining sufficient numbers of autologous T cells for CAR production from heavily pretreated, lymphopenic patients may pose a further problem, illustrating the clinical need for alternative CAR effector cell sources.

There is a rapidly growing interest in NK cells for CAR engineering due to their potent anti-tumor activity and safety in an allogeneic, “off-the-shelf” format which could overcome some of the limitations associated with autologous CAR T cell therapies. In a recent phase 1 and 2 trials, CD19-specific CAR-engineered primary NK cells have shown a tremendous clinical response in patients with relapsed or refractory CD19-positive non-Hodgkin’s lymphoma (NHL) or chronic lymphocytic leukemia (CLL) without the induction of typical CAR T cell-associated side effects such as cytokine release syndrome, neurotoxicity or graft-versus-host disease [14]. The majority of CAR NK cell studies to date, however, have been performed with NK-92 cells, a FDA-approved human cell line, which can be effectively expanded to high cell numbers and easily manufactured in a GMP-compliant manner. More importantly, early phase clinical trials have demonstrated the safety of irradiated NK-92 cells as an allogeneic cell therapeutic in patients with advanced hematological malignancies and solid tumors [15,16,17]. These properties make NK-92 cells an interesting option for CAR engineering and the development of standardized “off-the-shelf“ cell products with enhanced antitumor activity for adoptive cancer immunotherapy [18]. Despite the progress in treating hematological malignancies, CAR T cells in patients with solid tumors have demonstrated only limited antitumor activity [19]. Antigen escape is a key barrier for expanding the use of CAR-modified immune effector cells towards solid cancers with their more diverse surface antigen repertoires which are likely to fail single-targeted CAR therapy [20].

Since antigen heterogeneity and phenotypic plasticity of tumor cells present additional obstacles to the current development of CAR-based immunotherapies, efforts are being made to boost flexibility and improve the effectiveness by engineering modular chimeric antigen receptors so that the antigen recognition domain is split from the signaling domain of a conventional CAR, hence the target antigen can be switched or re-directed more readily without the requirement of re-engineering the CAR-modified immune effector cells [21].

We previously established a modular adapter CAR (AdCAR) platform which consists of AdCAR-expressing NK-92 cells that cannot recognize target antigens directly but are redirected to a target structure referred to as linker-label epitope, which consists of the endogenous vitamin biotin, conjugated to an adapter molecule (AM), e.g., a monoclonal antibody, in the context of special linker moiety, thereby allowing an on/off switch of CAR activity, and facilitating flexible targeting of various tumor antigens depending on the presence and specificity of the biotinylated AM [22,23,24,25,26]. Since novel or preexisting therapeutic antibodies can be easily labeled with biotin, there are almost unlimited possibilities in tumor antigen targeting using the AdCAR technology.

Bringing together the advantages of NK-92 cells as an “off-the-shelf” therapeutic and the controllable multiplex targeting capacity of the AdCAR system, led to the generation of a universal, on-demand CAR NK product which can be maintained and expanded at low cost in a GMP compliant manner for clinical use. Here, we outline the preclinical approach using AdCAR NK-92 cells in combination with therapeutic antibodies for targeting and elimination of bone metastatic cells in vitro using newly established bone metastatic cancer cell lines from different tumor entities, including mammary carcinoma, colorectal carcinoma, renal cell carcinoma and melanoma.

## 2. Results

### 2.1. Establishment and Characterization of Newly Developed Bone Metastasis Cell Lines

Tumor material from resected bone metastases and blood samples of the patients were provided by the Department of Orthopedic Surgery, University Hospital Tuebingen (UKT) (Table 1).

Cell lines were established from outgrowth cultures and successfully cultivated for more than 30 passages. All assays conducted throughout the present study were performed with cell lines in early passages between 5 and 7 to prevent potential, cell culture-induced mutational changes. Cell line authentication was performed by Eurofins Scientific using short tandem repeat (STR) analysis. Tumor cells were immunophenotyped for extracellular expression of tumor antigens that can be targeted by therapeutic antibodies using flow cytometry (Figure 1 and Table 2).

All cell lines shared high expression of B7-H3 (CD276), a recently emerging immune checkpoint molecule of the B7 superfamily. The epithelial growth factor receptor (EGFR), an important oncogene in the development of lung and colorectal cancer, is expressed in the mammary carcinoma MAC, the renal cell carcinoma MAM and the colorectal carcinoma MCK83 but not in the melanoma MeGa17. Interestingly, the melanoma cell adhesion molecule MCAM (CD146) was not only expressed in the MeGa17 cell line but also in the MAM cell line.

Further characterization of the tumor cell lines was conducted by flow cytometric assessment of cell surface expression of well-known NK cell ligands (Table 3). The colorectal carcinoma MAC, renal cell carcinoma MAM and melanoma MeGa17 showed comparable NK ligand expression profiles. Uniformly, all cell lines expressed human leukocyte antigen E (HLA-E), a major ligand for the inhibitory receptor complex CD94/NKG2A on NK cells, as well as the major histocompatibility complex (MHC) class I chain-related protein A and B (MICA/B), a protein that acts as an activating signal for NK cells through the natural killer group 2, member D (NKG2D or CD314) receptor. Moreover, all four cell lines expressed death receptor 5 (CD262), Nectin 2 (CD112) and PVR (CD155). CD262 is a cell surface receptor of the TNF-receptor superfamily that binds the tumor necrosis factor (TNF)-related apoptosis-inducing ligand (TRAIL) and mediates apoptosis. CD262 and CD112 are important ligands for DNAM-1 (CD226) triggering activating signaling cascades. They can also trigger an inhibitory NK response by activation of the immune checkpoint receptor TIGIT. Interestingly, the NK ligand HLA-ABC was completely absent on MCK83 cells while MAC, MAM and MeGa17 cells expressed it widely.

### 2.2. AdCAR NK-92 Cells Specifically Lyse Bone Metastasis Cell Lines In Vitro

NK-92 cells were transduced with lentiviral vectors encoding the second generation adapter CAR with a CD28 co-signaling domain and an intracellular immunoreceptor tyrosine-based activation motif (ITAM) from the CD3 zeta chain (CD3ζ). Cells were subsequently single-cell sorted for highest CAR expression and functionality. Transduction process and AdCAR NK-92 cell characterization was previously described [23]. AdCAR NK-92 cells showed stable AdCAR surface expression for at least 150 days after cell sorting with an average viability of >90%. Proliferation rate was not impaired by the transduction process compared to untransduced, parental NK-92 cells.

Functional assessment of AdCAR NK-92-mediated cytotoxicity was conducted using the previously established metastatic tumor cell lines earlier described in this manuscript as target cells. Calcein-labeled tumor cells were co-incubated with AdCAR NK-92 as well as parental NK-92 cells in the presence and/or absence of biotinylated antibodies (bAb) targeting antigens highly expressed on the tumor cells. AdCAR NK-92 but not untransduced NK-92 cells significantly induced cellular cytotoxicity against all tumor cell lines but only in the presence of a bAb targeting antigens sufficiently expressed on the cell surface (Figure 2a–d), thus, underscoring the controllability of the AdCAR system. Correlation of median fluorescence index (MFI) as an indicator of antigen density on tumor cell surface and specific cell lysis of MAC, MAM, MCK83 and MeGa17 cells resulted in R^2^ values of 0.5157, 0.1161, 0.7650 and 0.5375, respectively.

Next, we examined the kinetics of AdCAR-mediated cytotoxicity after addition of specific biotinylated antibodies. Utilizing the xCELLigence real-time label-free live cell analysis (RTCA) system based in cell impedance measurement, tumor cells were co-incubated with AdCAR and parental NK-92 cells with and without bAb and monitored for over 12 h. The dimensionless cell index is proportional to the amount of live tumor cells. NK-mediated cytotoxicity is assessed by measurement of cell index decrease. AdCAR NK-92 cells but not parental NK-92 cells successfully lysed the tumor cells of renal cell carcinoma MAM and melanoma MeGa17 in less than 4 h, but only in the presence of a specific bAb (Figure 3a,b). Specific tumor cell lysis correlated with surface expression of the respective antigen and no long-term tumor regrowth was observed with adapter molecules targeting highly expressed antigens.

To further examine NK-92-mediated lysis, a cytokine secretion profile was established to screen for secretion of a variety of cytokines, including NK cell effector molecules. Various cytokines were significantly increased after co-incubation of AdCAR-transduced NK-92 cells with MAC cells (Figure 4). GM-CSF (22-fold; *p* < 0.002), IL-10 (10-fold, *p* < 0.0002), granulysin (24-fold; *p* < 0.0006), granzyme B (6-fold, *p* < 0.0001), IFN-γ (10-fold; *p* < 0.0009), MIP-1b (2-fold; *p* < 0.008) and TNF-α (32-fold; *p* < 0.0001) showed significantly elevated levels but only upon AdCAR induction via specific biotinylated antibodies. While enhanced secretion of granulysin and granzyme B directly account for increased tumor lysis, IFN-γ and TNF-α stimulate the endogenous immune system and indirectly enhance anti-tumor activity. Secretion of MCP-1 and perforin was not significantly augmented after AdCAR activation (1.7-fold and 1.4-fold, respectively).

### 2.3. NK-92 Cells Exhibit Successful AdCAR-Mediated Cytotoxicity in a Three-Dimensional Tumor Cell Model

While the majority of in vitro studies about cellular immunotherapy are still based on tumor cell monolayer culture systems, examination of three-dimensional (3D) tumor models allows for limited translation to the in vivo situation. Thus, we generated multicellular spheroids of the GFP-transduced cell lines. Since just one out of the four cell lines successfully grew as a solid spheroid, only the renal cell carcinoma MAM was used to assess cytotoxic potential of AdCAR NK-92 cells in a 3D model.

After four days of culture tumor spheroids were co-incubated with either AdCAR-transduced or parental NK-92 cells in the presence or absence of biotinylated antibodies and monitored for over 96 h. Fluorescence signals of MAM spheroids co-incubated with NK-92 cells were correlated with untreated control spheroids (Figure 5a,b). After 48 h AdCAR NK-92 cells in combination with bCD146, bCD276 or bEGFR successfully increased NK-mediated lysis of MAM tumor cells to 76.9%, 81.1% and 80.3%, while AdCAR NK-92 cells in combination with bCD171 showed specific lysis of 51.4% of tumor cells. After 96 h AdCAR-mediated cell lysis increased to 82.3% (bCD146), 57.0% (bCD171), 83.5% (bCD276), 93.3% (bEGFR).

## 3. Discussion

To date, conventional therapies have limited success in preventing or treating bone metastasis due to the complex nature of the bone microenvironment, tumor heterogeneity, and the therapeutic resistance of dormant tumor cells. Despite significant advancement of conventional therapies, such as surgery, chemotherapy, hormone or radio therapy, metastatic disease remains virtually incurable and still is one of the most common causes for cancer-associated mortality [2].

Our recently developed AdCAR platform combines a modular adapter chimeric antigen receptor with the universal applicability of the NK-92 cell line, thus creating an “off-the-shelf” cellular product for the treatment of cancer [23]. Currently, universal CAR approaches with T or NK cells using adapter molecules that are foreign to the human body like the affinity-enhanced monomeric streptavidin 2 (mSA2) can potentially cause immunogenic reactions and make clinical translation difficult [27,28]. Even adapter molecules tagged with endogenous molecules such as biotin could potentially cause adverse reactions due to the immunogenicity of avidin and streptavidin [29]. The adapter system used herein is based on a scFv targeting a “neo”-epitope-like structure, the linker label epitope, consisting of biotin in the context of a mAb, instead of biotin itself [26]. Hence, application of AdCAR NK-92 cells can circumvent adverse immunogenic reactions of other CAR systems and decrease the risk of severe side effects during therapy.

The flexibility of the presented AdCAR system can further counteract drawbacks of conventional CAR T cell therapy such as off-tumor toxicity or tumor evasion strategies. Previous CAR T cell studies showed downregulation of the target antigen as a reaction to therapy which ultimately leads to immunotherapy resistance [30,31,32]. Additionally, since only few antigens are uniquely specific for solid tumors, eradication of tumor cells while simultaneously sparing healthy tissue is a major concern of CAR T cell therapy [33]. Utilizing a standardized flow cytometry screening panel enables target options that are tailored on a patient-individualized basis. In case of tumor antigen evasion during therapy, the target structure can be easily switched by application of a different bAb while still retaining therapeutic efficacy. Moreover, simultaneous or consecutive use of different biotinylated antibodies in combination with AdCAR NK-92 cells may provide a possible treatment strategy for highly heterogeneous tumors and is able to counteract tumor antigen loss and off-tumor/on-target toxicity. Utilization of biotinylated antibodies, which are already approved by the FDA, such as cetuximab or trastuzumab generates functional adapter molecules with a known safety profile for AdCAR NK-92 therapy and facilitates translation into clinical settings.

In this study, we demonstrated that CAR-modification of the NK-92 cell line with our AdCAR system can enable promising therapeutic opportunities for the treatment of a variety of metastatic tumor entities. AdCAR NK-92 cells effectively eliminated tumor cells from the newly established cell lines derived from bone metastases of renal cell, mammary and colorectal carcinomas, as well as melanomas in standard in vitro cytotoxicity assays within two hours. Specificity of cytolytic activity was achieved by prior screening for suitable target antigen structures on tumor cell lines and utilization of respective biotinylated antibodies as adapter molecules. Due to the lack of FcγRIII (CD16) expression on NK-92 cells, the formation of the immunological synapse is simply dependent on bAb titration [34]. AdCAR NK-92 cells that were co-incubated without bAb or with bAb without specificity of interest lead to little to no tumor lysis. Additional controllability of AdCAR NK-mediated cytotoxicity is underscored by the fact that cytolytic activity occurred in a concentration-dependent manner [23]. Moreover, AdCAR NK-92 cells are functionally independent of the target cells’ NK ligand profile. Specific AdCAR-mediated cytotoxicity could be demonstrated regardless of tumor cell surface expression of inhibitory ligands such as HLA-E which was shown to have negative effects cytotoxic activity of primary NK cells [35,36].

As shown in first in vivo and clinical CAR NK-92 trials the NK cell line was less likely to induce severe side effects such as neurotoxicity and cytokine release syndrome (CRS) [37,38]. Likewise, in the present study, IL-6, a driving causation of CRS, was not produced by AdCAR NK-92 cells after co-incubation with either tumor cell line or a specific bAb. The secretion of the pro-inflammatory cytokines IFN-γ and TNF-α, as shown by AdCAR NK-92 cells co-incubated with the target cell line and a specific bAb, may, additionally, stimulate the endogenous immune system and enhance anti-tumor activity [18].

One of the key necessities of CAR-immune cell therapies to be successful for the treatment of solid tumors and, especially, in the metastatic setting is their capability of immune cell homing and tumor infiltration [39,40,41]. Establishment of three-dimensional in vitro spheroids enables accurate assessment of a range of in vivo biological processes [42]. Here, AdCAR NK-92 cells are clearly able to efficiently lyse tumor spheroids in an antibody-dependent manner. Furthermore, they have been shown to be resistant to the tumor microenvironmental influences such as the secretion of transforming growth factor beta (TGFβ) which was reported to account for tumor resistance to immunotherapy [18,43,44]. Future research needs to evaluate the therapeutic potential of AdCAR NK-92 cells in pre-clinical in vivo and clinical settings. Particularly, mouse bone metastasis models will be of major importance to assess AdCAR NK-92 biodistribution, homing and infiltration.

As previously described, frozen CAR-engineered NK-92 cells can be thawed and expanded as a batch culture in gas permeable cell culture bags. Doubling time of CAR NK-92 cells is approximately 32 to 36 h and culture yields can be individually scaled according to treatment doses and number of patients treated [45]. Safety was proven in a recent phase I clinical trial with CD33-specific CAR NK-92 cells. No dose-limiting toxicities were observed upon repeated intravenous infusions of up to 5 × 10^9^ irradiated cells per dose [37]. Several other early phase clinical trials with CAR-engineered NK-92 cells are also currently being carried out in Europe, China and the US (clinicaltrials.gov; NCT 02742727, 03383978, 04050709).

For clinical translation, metastatic tumor cells isolated from bone marrow biopsies or surgery can be quickly screened for their target antigen expression profile with the implemented, standardized antibody panel using flow cytometry or ultra-high content imaging techniques. AdCAR-engineered NK-92 cells could be utilized to eradicate disseminated, dormant metastatic tumor cells within the bone marrow as well as micrometastases. Furthermore, for the treatment of large bone metastases, AdCAR NK-92 cells could be given as intratumoral injections or also be applied after surgery at the resection site to eliminate non-resectable tumor residues. A well-characterized, GMP-compliant qualified master cell bank of CAR NK-92 cells as a reliable source for subsequent production of patient doses, enables therapeutic translation within hours of the surgery [38,46]. Together with biotinylated therapeutic antibodies, universal “off-the-shelf” AdCAR NK-92 cells can be utilized for flexible and patient-individualized therapy, thus, enabling broadly available and more affordable immunotherapy for cancer patients independent of specialized facilities. Their specific properties make AdCAR NK-92 cells a promising treatment option for bone metastases.

## 4. Materials and Methods

### 4.1. Cell lines and Culturing Conditions

NK-92 cells were purchased from ATCC and maintained at a concentration of 10^5^ cells/mL in MEM Alpha Medium containing stable L-glutamine (GlutaMAX, Thermo Fisher Scientific, Waltham, MA, USA) supplemented with 20% FBS and 100 U/mL IL 2 (Proleukin, Aldesleukin, Chiron, Emeryville, CA, USA) referred to as NK-92 complete medium. MAC, MAM, MCK83 and MeGa17 cell lines were established from biopsy tumor samples from resected bone metastases provided by the Department of Orthopedic Surgery, University Hospital Tuebingen (UKT). Newly established cell lines and matched tumor samples, as well as patient’s blood lymphocytes for comparison, were authenticated by short-tandem-repeat analysis (Eurofins Scientific, Luxembourg City, Luxembourg) to verify cell origin and identity. Tumor cell lines were cultivated in RPMI 1640 medium (Thermo Fisher Scientific, Waltham, MA, USA) supplemented with 10% heat-inactivated fetal bovine serum (FBS) (Thermo Fisher Scientific, Waltham, MA, USA) containing GlutaMAX, referred to as RPMI complete medium, maintained at 37 °C in a humidified 5% CO_2_ atmosphere and regularly tested for mycoplasma contamination. The present study was approved by the ethics committee at the Medical Faculty of the Eberhard Karls University and the University Hospital Tuebingen (reference number 008/20114/BO2). Human material was collected after obtaining informed consent in accordance with the Helsinki protocol. All media contained 1x antibiotic-antimycotic solution (Thermo Fisher Scientific, Waltham, MA, USA).

### 4.2. Design of the AdCAR System

The second-generation adapter CAR is based on the mAb “mBio3”-derived single-chain variable fragment (scFv) targeting a “neo”-epitope-like structure, the linker label epitope, consisting of biotin in the context of a mAb. Exact constitution of the AdCAR construct and generation of AdCAR-engineered NK-92 cells was previously described [23].

### 4.3. Biotinylated Antibodies

Antibodies were either purchased as bAb from Miltenyi Biotec (Bergisch-Gladbach, Germany) or acquired from the UKT pharmacy and biotinylated by Davids Biotechnologie (Regensburg, Germany).
**Antigen****Clone****Antibody****Order #****Lot****Supplier**CD146541-10B2n/a130-092-8525190627154Miltenyi Biotec, Bergisch Gladbach, GermanyCD171REA163n/a130-100-7025190607129Miltenyi Biotec, Bergisch Gladbach, GermanyCD200OX-104n/a130-106-0645191021606Miltenyi Biotec, Bergisch Gladbach, GermanyCD221REA271n/a130-103-9735190627184Miltenyi Biotec, Bergisch Gladbach, GermanyCD271REA844n/a130-112-6085190627191Miltenyi Biotec, Bergisch Gladbach, GermanyCD274n/aAtezolizumabn/an/aHoffmann-La Roche, Basel, SwitzerlandCD276FM276n/a130-095-5145190627174Miltenyi Biotec, Bergisch Gladbach, GermanyCD340n/aTrastuzumabn/an/aHoffmann-La Roche, Basel, SwitzerlandEGFRn/aCetuximabn/an/aMerck KgaA, Darmstadt, GermanyGD2n/aDinutuximab betan/an/aEusa Pharma, Hertfordshire, Great Britain


### 4.4. Flow Cytometry

Staining of cells was conducted using primary biotinylated mAb with antigen specificity of interest. Cells were incubated with specific antibodies at 4 °C in flow cytometry buffer containing PBS (Sigma-Aldrich, St. Louis, MO, USA) supplemented with 2% FBS and 0.5 M EDTA (Sigma-Aldrich, St. Louis, MO, USA) for 15 min. Unbound antibody was washed off by centrifugation (4 °C, 350 g, 5 min) and stained with a secondary, fluorophore-labeled anti-biotin antibody (Anti-Biotin PE, clone: REA746, Miltenyi Biotec, Bergisch Gladbach, Germany) for 15 min followed by another washing step. Surface antigen expression was analyzed with the secondary antibody alone as a negative control using a BD FACSCanto II flow cytometer (BD, Franklin Lakes, NJ, USA).

### 4.5. Calcein Release-Based Cytotoxicity Assay (CRA)

Target cell staining with Calcein AM (Thermo Fisher Scientific, Waltham, MA, USA) as well as the protocol for the calcein release-based cytotoxicity assay (CRA) was described previously [23,45].

### 4.6. Real-Time Label-Free Live Cell Analysis

Bone metastasis cell lines were adjusted to a concentration of 10^5^ cells/mL in RPMI complete medium and seeded in E-Plate 96 VIEW (OLS, Bremen, Germany) micro-well plates. Effector AdCAR NK-92 cells were adjusted to an E:T ratio of 5:1 in NK-92 complete medium without IL-2 and co-incubated with the target cells in the presence or absence of specific bAb. Utilizing the xCELLigence real-time cell analysis (RTCA, OLS, Bremen, Germany) system, cells were monitored for over 12 h. Tumor cell viability was calculated using the RTCA 2.0 software (OLS, Bremen, Germany) and AdCAR-mediated cytotoxicity was subsequently determined.

### 4.7. Quantification of Cytokine Release

Cytokine release of AdCAR NK-92 cells upon AdCAR induction was determined using the Bio-Plex Pro human cytokine 17-plex assay (Bio Rad, Hercules, CA, USA). The respective protocol was described previously [23].

### 4.8. 3D Spheroid Cytotoxicity Assay

GFP-transduced metastatic tumor cells were grown as three-dimensional spheroids and co-incubated with NK-92 as well as AdCAR NK-92 cells in the presence or absence of biotinylated antibodies. The respective protocol was described previously [23].

### 4.9. Data Analysis

All statistical analyses were performed with GraphPad Prism 8 software (GraphPad Software Inc., San Diego, CA, USA). Flow cytometry data were analyzed using FlowJo software V10.0.8 (FlowJo LLC, Ashford, OR, USA).

## 5. Conclusions

Innovative immunotherapy for the treatment of solid cancers and especially metastatic disease is urgently needed. Adapter CAR-engineered NK-92 cells are able to combine their “off-the-shelf” availability with personalized and controllable elimination of metastatic tumor cells, thus, establishing a potent cellular product with universal applicability and quick clinical translation potential for the treatment of solid tumors, including metastases.

## Figures and Tables

**Figure 1 cancers-13-01124-f001:**
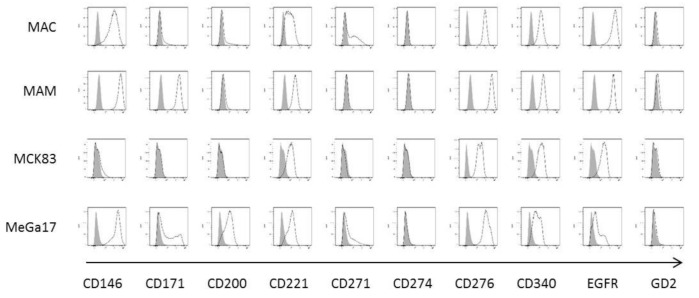
Cell line immunophenotyping. Bone metastatic cell lines were screened for tumor antigen surface expression using flow cytometry. Cells were co-incubated with primary biotinylated antibodies for 15 min at 4 °C. Antigen expression was detected using a secondary PE-coupled anti-biotin antibody (black line) using staining with the secondary antibody alone as negative control (grey area).

**Figure 2 cancers-13-01124-f002:**
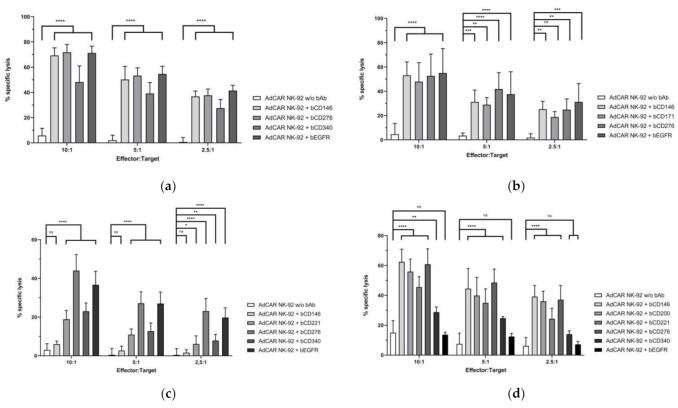
AdCAR NK-92-mediated metastatic tumor cell lysis. AdCAR NK-92 cells were co-incubated with calcein-labeled tumor cell lines MAC (**a**), MAM (**b**), MCK83 (**c**) and MeGa17 (**d**) in the presence or absence of indicated biotinylated antibodies for 2 h at indicated E:T ratios. Specific lysis is shown as mean ± SD, n = 3. ****: *p* < 0.0001; ***: *p* < 0.001; **: *p* < 0.01; *: *p* < 0.1; ns: *p* ≥ 0.1.

**Figure 3 cancers-13-01124-f003:**
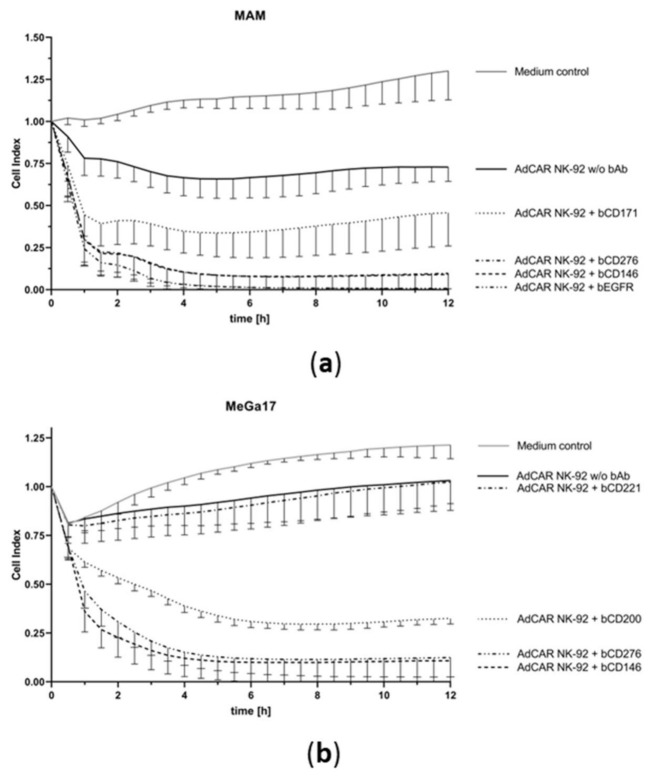
Kinetics of AdCAR-mediated tumor cell lysis. AdCAR NK-92 cells were co incubated with unlabeled tumor cell lines MAM (**a**) and MeGa17 (**b**) in the presence or absence of indicated biotinylated antibodies and constantly monitored over time using the xCELLigence real time cell analysis system. NK-mediated tumor cell lysis is depicted as decrease in the dimensionless “cell index”, *n* = 3.

**Figure 4 cancers-13-01124-f004:**
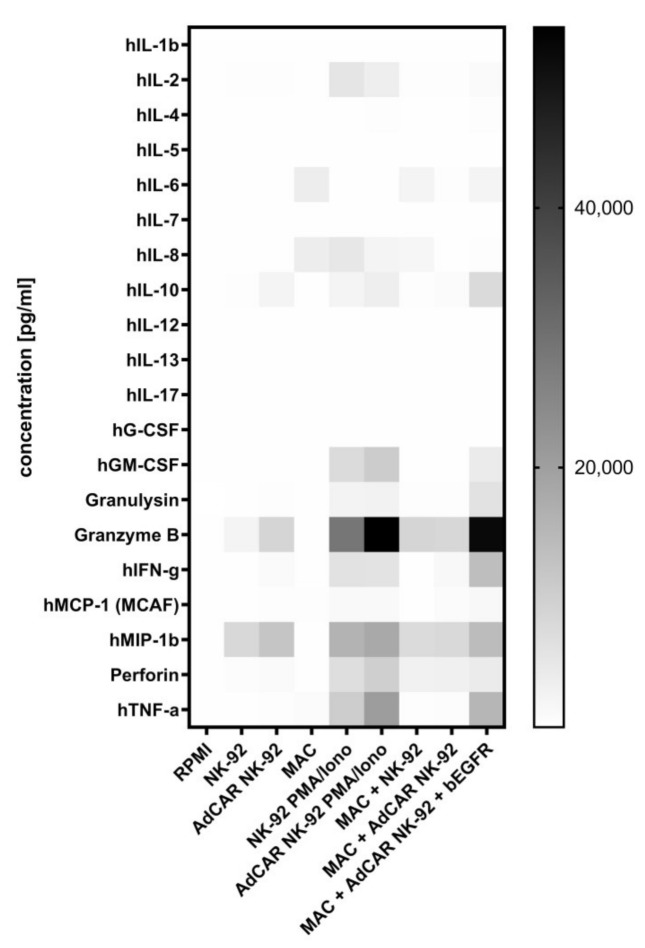
Cytokine secretion profile of AdCAR NK-92 cells. AdCAR NK-92 cells as well as parental NK-92 cells were co-incubated with the tumor cell line MAC in the presence and absence of bEGFR for 6 h at an E:T ratio oif 5:1. The release of cytokines was measured using the Bio-Plex Pro human cytokine 17-plex assay and is shown as a heatmap. PMA/Ionomycin was used as control to induce maximum cytokine secretion.

**Figure 5 cancers-13-01124-f005:**
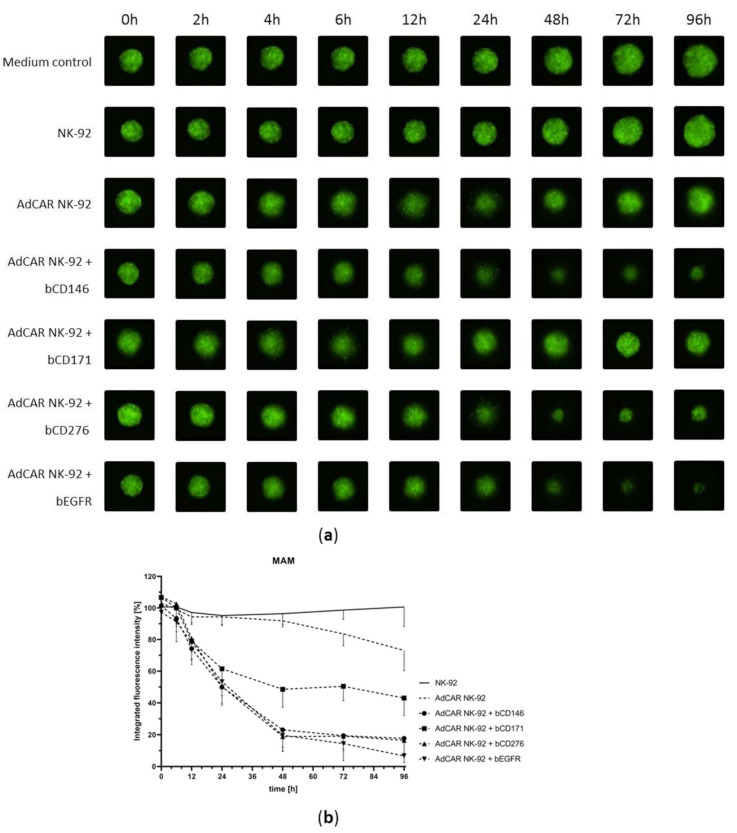
AdCAR NK-92-mediated lysis of 3D tumor spheroids. GFP-transduced cell lines MAM was grown as 3D spheroids in ultra-low attachment plates and subsequently co-incubated with AdCAR NK-92 or parental NK-92 cells and indicated biotinylated antibodies for 96 h in at least three individual experiments. Fluorescence images show representative MAM spheroids at indicated time points (**a**). Integrated fluorescence intensity of tumor spheroids was measured regularly using the Celigo S Imaging Cytometer (Nexcelom, Lawrence, MA, USA), compared to untreated control spheroids and is shown as mean ± SD, *n* = 3 (**b**).

**Table 1 cancers-13-01124-t001:** Patient data for tumor cells from bone metastasis resections.

	Patient Age [years]	Patient Sex	Metastatic Site	Tumor Entity	Designation
**Patient 1**	63	f	Scapula	Mammary carcinoma	**MAC**
**Patient 2**	17	m	Spine L3/L4	Renal cell carcinoma	**MAM**
**Patient 3**	54	m	Spine C7	Colorectal carcinoma	**MCK83**
**Patient 4**	47	m	Acetabulum	Melanoma	**MeGa17**

**Table 2 cancers-13-01124-t002:** Cell line immunophentoyping. Tumor cells were co-incubated with primary biotinylated antibodies. Antigen expression was detected using a secondary PE-coupled anti-biotin antibody. Percentage of stained cells and median fluorescence index (MFI) values were calculated using staining with the secondary antibody alone as negative control.

**Cell Line**	**Percentage of Stained Cells**									
	**CD146**	**CD171**	**CD200**	**CD221**	**CD271**	**CD274**	**CD276**	**CD340**	**EGFR**	**GD2**
**MAC**	97.70%	11.80%	13.90%	68.50%	44.60%	0.40%	100.00%	97.90%	99.90%	1.78%
**MAM**	99.80%	99.90%	5.29%	99.40%	2.06%	0.65%	99.90%	99.70%	100.00%	4.04%
**MCK83**	5.62%	0.45%	0.31%	59.60%	0.13%	0.46%	99.50%	81.20%	85.80%	0.44%
**MeGa17**	99.20%	42.90%	62.10%	70.40%	22.80%	1.23%	99.00%	44.10%	27.20%	1.91%
	**Median Fluorescence Index (MFI)**									
	**CD146**	**CD171**	**CD200**	**CD221**	**CD271**	**CD274**	**CD276**	**CD340**	**EGFR**	**GD2**
**MAC**	45.48	1.15	1.23	4.10	1.91	1.03	106.65	9.39	183.42	1.21
**MAM**	194.20	79.54	1.16	11.27	0.99	1.03	113.92	11.59	92.86	1.35
**MCK83**	1.22	1.00	0.96	6.26	0.77	0.97	31.39	10.08	12.41	1.14
**MeGa17**	172.52	3.49	7.71	9.06	1.96	0.96	67.70	5.00	2.78	1.18

**Table 3 cancers-13-01124-t003:** Characterization of tumor cell lines for NK ligand expression. Tumor cells were co-incubated with fluorescently labeled antibodies. MFI values were calculated using staining with the respective isotype control antibody.

**Cell Line**	**CD48**	**CD50**	**CD54**	**CD58**	**CD95**	**CD102**	**CD112**	**CD155**	**CD261**	**CD262**
**MAC**	1.47	1.69	5.46	73.22	23.35	1.27	34.41	128.96	3.01	16.87
**MAM**	1.36	1.78	7.31	30.55	9.54	1.13	69.73	192.65	6.06	59.84
**MCK83**	1.84	1.75	1.29	5.54	4.26	1.27	15.32	89.62	4.15	14.91
**MeGa17**	1.25	1.58	9.66	31.32	2.18	1.13	16.27	50.20	1.22	15.03
	**HLA-ABC**		**HLA-DR**	**HLA-E**	**HLA-G**	**MICA/B**	**ULBP1**	**ULBP2/5/6**	**ULBP3**	**ULBP4**
**MAC**	52.41		0.90	9.18	0.98	6.53	3.45	1.26	0.68	0.75
**MAM**	19.07		0.87	8.07	0.81	21.25	3.15	1.00	1.67	1.53
**MCK83**	0.47		0.92	10.01	0.82	9.07	6.09	1.09	1.25	1.41
**MeGa17**	23.33		12.10	7.55	1.34	7.27	2.45	1.37	0.92	0.77

## Data Availability

The data presented in this study are available from the corresponding author upon reasonable request.

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
