# Peer review of "Adapter Chimeric Antigen Receptor (AdCAR)-Engineered NK-92 Cells for the Multiplex Targeting of Bone Metastases"

_cancers, 2021, doi:10.3390/cancers13051124_

Round 1

Reviewer 1 Report

Grote et al present results characterizing a natural killer cell line (NK92) genetically modified to express an adapter chimeric antigen receptor (AdCAR) for the targeting of clinically relevant bone metastases. The authors present data demonstrating that these cells can be selectively targeted using commercially available and well characterized antibodies to kill cell lines established from the metastatic lesions when grown in monolayers or spheroids.

Major limitations:

1. This work is a direct extension of previously published and cited work by the same group, it does not in this reviewers opinion provide much new information or establish further clinical relevance beyond the suitability of potentially relevant antibodies for bone metastases. Xenograft models seem like an appropriate next step to determine whether this promising combination cell therapy will be beneficial in vivo. This or similar approaches that provide further evidence for this approach would be welcomed.  

Reviewer 2 Report

Summary: Targetted immunotherapy is needed so individuals do not suffer from metastatic spreading of solid tumor cells. AdCAR NK-92 mediated cytotoxicity in vitro was done in 2D and 3D models to assess against four different tumors that were found metastasized to bone. ADCARNK92 cells produced a lot of NK effector molecules and pro inflammatorycytokines, and had increase cytotoxicity in 3D models . These cells are a promising cellular product for targeted therapy of cancers metastasizing to the bone.

  1. Introduction
    1. The introduction is quite long. Given the focus on the paper is NK cells, would it be possible to trim down some of the background information regarding bone metastases and expand ever so slightly on the NK cell descriptions?
  2. Results
    1. Line 155 states that cell lines were established from outgrowth cultures and cultivated for more than 30 passages. Which passages were used for analyses? Were passages normalized throughout the duration of the experiment and kept consistent between analyses? It’s very impressive that the authors were able to successfully grow these primary tumor lines, but I am concerned about potential changes in the cells during analyses due to passage differences.
    2. Tables 2 and 3 show the very impressive phenotypic profile that the authors conducted on the tumor cells.
    3. Line 197 – what was the threshold for highest CAR expression and functionality on NK-92 cells?
    4. Line 214 – why was median fluorescence index used when mean fluorescence index was use in the previous section?
    5. What was the rationale for a 12 hour incubation of AdCAR NK-92 with and without bAb?
    6. How was NK cytokine secretion assessed to exclude possible secretion from tumor cells?
    7. What was the rationale for 4 day 3D culture (as well as the unique timeline for this culturing?)
  3. Discussion
    1. Discussion is very well written and I appreciate the ability of the authors to discuss these data in an easy way.
    2. It would be nice if the authors could discuss how application of this technique could be used in the clinic, and potential ways it might be feasible (if possible, especially given that this is all in vivo and would require lots of research investigating the safety and efficacy in vivo as well).
    3. Can the authors describe any other cancers that might be able to be controlled using this? Can this therapy be adapted to unique types of cancers in individuals – thereby making it individualized immunotherapy? What is the timeline of all of these assays and investigation that could potentially make this protocol something used in the clinic?
    4. Though bone is very intricately connected with the vasculature, how likely is it that adCAR-NK-92 cells would mobilize to the mets of the cancer? Can the authors describe how this might be done and how this therapy could be used in clinic?

Reviewer 3 Report

The manuscript entitled “Adapter chimeric antigen receptor (AdCAR)-engineered NK-92 cells for the multiplex targeting of bone metastases”, the authors evaluated AdCAR NK-92 cell efficacy as a possible treatment strategy for various types of bone metastatic cancers.

In this study, patient-derived bone metastatic tumor cells were established as cell lines, the expression of surface target ligands were analyzed, and it was confirmed that they exhibited cytotoxicity specifically for each antigen of cancer cells using approved adapter molecules. Although functionality of AdCAR NK-92 as a universal CAR NK cell was shown, this manuscript is somewhat insufficient to present the preclinical efficacy evaluation and treatment strategies for tumors that the authors intended.

As in this study, when patient-derived tumor cells are cultured for more than 30 passages and established as a cell line, some of the cellular characteristics in the early passage are often lost. In adhesion-based cell culture, there may be a problem in that cell types with a higher adhesion among heterogeneous cells are predominantly cultured.

This manuscript needs to be amended according to the purpose/constitution of the study.

  1. If the development of patient-individualized/targeted immunotherapy is the primary focus of the study, the use of tumor cells in early passage and cultures containing suspended or low adherent cells would be useful in developing patient-individualized therapies and better representative of patient cancer cells.

Refer to “Switchable CAR-T cells mediate remission in metastatic pancreatic ductal adenocarcinoma”.

  1. On the other hand, if the purpose is to identify the characteristics and specific markers of bone metastatic tumor cell lines, Figure 1 and Table 2 should be reconstructed with major common surface antigens (ex> CD146, CD171, CD200, CD221, CD276, CD340, EGFR) from established cell lines, and additional explanations for each antigen are needed. In addition, the expression pattern of CD146 in the MeGa17 cell line in Figure 1 is similar to that of the negative control, but the MFI value of CD146 in Table 2 is 172.52, which is not well matched. Therefore, confirmation of the MFI value is required, and the percentage of antigen expression compared to the negative control should be marked in parallel considering the difference in cell line. In Table 3, it would be more appropriate to show the expression patterns of NK ligands in cancer cell lines as supplementary data, and additional experiments are required to show the relationship between the efficacy of AdCAR NK-92 cells and expression of NK ligands.
  2. If the treatment strategy for AdCAR NK-92 cells is the objective, a new treatment and administration method for AdCAR NK-92 cells should be suggested. Rather than single targeting with one adapter molecule against a single antigen, a multi-targeting strategy that applies multiple adapter molecules against several antigens identified in bone metastatic tumor cell lines would be more efficient. Moreover, in this manuscript, the efficacy of AdCAR NK-92 cells was evaluated only by in vitro assay. The in vitro assay has a significant disadvantage in that it can not reflect the tissue barrier and circulation of AdCAR NK-92. Therefore, it is possible to establish a more practical preclinical efficacy test and administration strategy of AdCAR NK-92 cells and adapter molecules in in vivo bone metastasis model.

Questionable points of the manuscript are how clear in the direction of the study, and how well organized in terms of datum presentation/description, and rationale for the flow of experiments presented in each of the figures.

Reviewer 4 Report

In this article, Grote et al. described another application of the adapter chimeric antigen receptor gene-modified NK-92 with particular reference in the treatment of refractory oncological patients with bone metastases. Although the manuscript is very well written several criticisms need to be pointed out:

  1. The introduction is too long (2 pages) and dispersive. It should be more concise and define better the state of art of the technology and focus on key issues that the authors would like to address. The paragraph on CAR description is inaccurate (row 86-89). If the authors would like to describe it, they need to do it properly (not only CD3ζ is used but also the γ chain of light affinity IgE Fc receptor; then not only the mentioned costimulation molecules have been used to generate second and third generation of CARs – DAP10, CD244, MyD88_CD40, etc). I suggest to simply remodulate the sentence.
  2. It will be easier for a reader to have the study divided in paragraphs that highlight the results obtained.
  3. The figure 1 and the table 2 contradict each other and they do not reflect what is described in the text. It is confusing and not clear which cell line express what…
  4. No cell line characterization is reported in the text except for the identity (STR analysis, proving that they are derived from a specific patient). How are the authors sure that these cells are representative of the original tumors? Did the authors check for the expression of lineage markers and mutations (tumor molecular characteristic)? Are the cells homogeneous both in terms of dimension and antigens expression or are they heterogeneous?
  5. Regarding the AdCAR stability declared (row 199), how did the authors define it? Did the authors keep in consideration genome stability, integrity issues (mutations due to the nature of the cell line – tumor) and possible recombination events with the latent EBV expression (PMID: 20454443)?
  6. Regarding the functionality of the AdCAR NK-92:
    1. Did the authors perform the experiments using FBS free media recapitulating the GMP manufacturing process and proving the real efficiency of the generated cellular product?
    2. Since NK-92 are suggested as an off-the-shelf product, for their clinical application they need to be frozen and irradiated before their infusion because they are an immortalized leukemia cell line. Did the authors perform the functional experiments using frozen and irradiated AdCAR NK-92? Until which effector:target ratio is it possible to observe the efficacy of the AdCAR NK-92? How do the authors explain the low killing of MAM using the CD171 adapter although its expression is extremely high (third decade in the staining, Figure 1)?
    3. The data presented in Figure 5 are extremely nice however only one cell line has been analyzed. In the text or in the figure legend there is a mistake in the cell line used. Then, the authors declare a better infiltration, but no data are reported.
  7. In the discussion, the authors should point out several important aspects including:
    1. biodistribution of the Adapter,
    2. AdCAR homing, persistence and infiltration until Adapter is infused
    3. Clinical availability of the selected Adapter
    4. definition of the characteristic to make this cellular product available also in non-USA countries (safety, potency, etc…),
    5. include reference also of the use of third-party NK cell genetically modified with CAR (PMID:32023374),
    6. Remodulate the sentence in row 333: No infiltration data has been demonstrated.
  8. Row 349: modify 105 to 10e5

Round 2

Reviewer 4 Report

I would like to thanks the authors to address all my observations and comments.

I would like to invite them to further modify the list of clinical trials reported in row 336-337 and insert one for USA, China and Europe (the only one present in Germany). I do not know if the authors would like to include also the Canadian study using a non gene modified NK-92.